# GRAFTA: A GRAPH ATTENTIONAL FEW-SHOT AUDIO TAGGING NETWORK

## ABSTRACT

Audio tagging models based on CNNs and Transformers achieve strong results, but they rely on large labeled datasets and heavy computation, and struggle with the irregular structure of spectrograms. We propose GRAFTa, a Graph Attentional Few-shot Audio Tagging framework that represents spectrogram regions as nodes in a dynamically constructed graph, enabling attention to capture both local and long-range acoustic dependencies. To support multi-label generalization from few examples, GRAFTa constructs compositional subset prototypes: embeddings for both individual labels and frequently co-occurring label subsets mined from the support set. We evaluate GRAFTa in episodic few-shot settings on FSD50K (Fonseca et al., 2021) and AudioSet-Balanced, achieving 0.2296 and 0.2221 mean average precision (mAP), respectively. To our knowledge, this is the first work to establish benchmarks for few-shot multi-label audio tagging on these datasets, highlighting both the promise of graph-based compositional prototypes and the unique challenges of this setting.

## 1 INTRODUCTION

Understanding real-world audio requires reasoning over complex, overlapping sound events and generalizing beyond abundant supervision. Audio tagging—assigning class labels to audio clips—has advanced through CNNs and Transformers, learning robust mappings from spectrogram features to semantic labels.

Despite their success, existing models face three key challenges in real-world tagging scenarios: (1) they struggle with the irregular and non-local structure of spectrograms, (2) they assume ample labeled data per class, and (3) they often treat multi-label predictions as independent decisions, overlooking the compositional nature of acoustic events. CNNs are inherently local in scope, while Transformers typically flatten spectrograms into token sequences, losing spatial structure. Moreover, multi-label audio tagging poses a harder generalization problem than single-label classification, especially under few-shot constraints.

### 1.1 OUR APPROACH.

We propose **GRAFTa**, a **Gr**aph-**A**ttentional model for **F**ew-shot **T**agging of **A**udio. GRAFTa constructs a graph from the input log-mel spectrogram, treating time-frequency regions as nodes and modeling their relationships via a Graph Attention Network (GAT). This graph structure allows dynamic propagation of acoustic context across non-local regions, capturing both co-occurrence and global dependencies in a spatially structured way.

To improve generalization with limited labeled data, we use a prototypical learning framework that compares queries to class prototypes built from a small support set Rather than learning static class weights. For multi-label tagging, we extend this approach with compositional subset prototypes—capturing not only individual labels but also common label combinations discovered from the support examples. This allows the model to represent complex events (e.g., dog barking with traffic) with richer, more meaningful semantics.

We evaluate GRAFTa on FSD50K and AudioSet-Balanced in a few-shot setting, and, to our knowledge, provide the first few-shot multi-label benchmarks on these datasets, setting a baseline for future audio meta-learning research.

## 2 RELATED WORK

• In **Audio Tagging** Deep learning has advanced sound classification, primarily via CNNs that effectively model local spectrogram patterns Kong et al. (2020). However, CNNs often miss long-range dependencies; to address this, Transformer-based architectures Gong et al. (2021) have been explored for their superior global context modeling, though this typically comes at a higher computational cost

• In **Graph Neural Networks in Audio**, Graph neural networks (GNNs) are gaining traction in audio processing for their strength in handling non-Euclidean structured data, with successful applications in speech enhancement, music structure analysis (Chen et al., 2020), and event localization (Wang et al., 2022). and dynamic graph-based spectrogram modeling for sound classification (Lu et al., 2022). Still, few works explore GNNs under few-shot conditions or combine them with metric-based reasoning.

• In **Few-Shot Learning for Sound** Few-shot sound event recognition aims to identify new classes from very few labeled examples. Existing methods rely heavily on meta-learning Wang et al. (2020), transfer learning with pretraining Lin et al. (2021), and metric-based methods like prototypical networks (Snell et al., 2017). but most ignore the rich spatial and temporal structure of audio spectrograms. Our work bridges this gap by integrating graph attention over spectrogram regions with prototypical learning, yielding better generalization and higher data efficiency in low-shot settings.

• In **Multi-Label Few-Shot Audio Learning** Prior work has explored few-shot learning for multi-label audio classification, including approaches based on meta-learning (Cheng et al., 2019), taxonomy-guided learning (Liang et al., 2024), and continual learning frameworks (Wang et al., 2021). However, these methods typically employ a base/novel class split, where a large set of base classes (e.g., 100+ classes) is used for pretraining before evaluating on novel classes. Additionally, graph-based audio tagging approaches such as ATGNN (Singh et al., 2024) focus on fully-supervised settings with complete label availability. In contrast, our work addresses a more challenging setting: we perform episodic evaluation across the full label taxonomy (all 200 classes in FSD50K, all 527 in AudioSet) without base class pretraining, and we explicitly model label co-occurrence through compositional subset prototypes rather than learning static label dependencies. This makes direct numerical comparison infeasible without reimplementing prior methods under our protocol.

## 3 METHODOLOGY

In this section, we detail our data pipeline and the three main components of GRAFTa: spectrogram preparation, graph-based feature extraction, and prototype-based matching.

**Notation** We use the following notation throughout: $F = 1025$ (STFT frequency bins), $T$ (STFT time frames), $N = 33$ (temporal nodes after CNN), $D = 256$ (CNN feature dimension), $d' = 64$ (GAT per-head dimension), $H = 8$ (attention heads), $d_{\text{emb}} = 256$ (final embedding dimension), and $C$ (classes per episode).

### 3.1 DATA PREPARATION AND AUGMENTATION

**Spectrogram Conversion** Raw audio $x(t)$ is resampled to $f_s = 22{,}050$ Hz, converted to mono, and either trimmed or zero-padded to a fixed 10-second length. We compute the Short-Time Fourier Transform (STFT) with window size $n_{\text{FFT}} = 2048$ and hop length $h = 209$, yielding $X \in \mathbb{C}^{F \times T}$, where $F = 1025$ is the number of frequency bins and $T$ is the number of time frames. A fixed mel filterbank $\Phi \in \mathbb{R}^{128 \times F}$ projects the power spectrogram:

$$M = \Phi \, |X|^2, \quad M \in \mathbb{R}^{128 \times T}$$

The resulting mel spectrogram is converted to decibels and min–max normalized to $[0, 1]$.

**Subset Construction and Pruning**  For each episode, we enumerate all non-empty label subsets of size up to three from the current support set. This guarantees that singleton classes are always included (subsets of size one), while allowing the model to capture common co-occurrence patterns via size-2 and size-3 subsets. In practice, the total number of such subsets is small (e.g., at most $2^5 - 1 = 31$ in a 5-way episode). When the number of possible subsets exceeds 500, we retain the first 500 subsets after sorting by subset size and lexicographic order, providing computational efficiency while preserving the most fundamental label combinations.

**Data Augmentation**  At each training iteration, we apply two augmentation techniques:

- **Mixup:** We blend spectrograms and labels using:

$$\widetilde{M} = \lambda M + (1 - \lambda)M', \quad \widetilde{y} = \text{clamp}(y + y', 0, 1)$$

  where $\lambda \sim \text{Beta}(0.5, 0.5)$. We use element-wise clamping to ensure label values remain in $[0, 1]$ for the multi-label setting.
- **SpecAugment:** We randomly mask up to 15% of mel bins and 20% of time frames to improve generalization.

**Channel Replication and Standardization**  Each $128 \times T$ spectrogram is replicated across 3 channels to match the CNN14 backbone requirements. Finally, we apply standardization:

$$S = \frac{M_{\text{rep}} - \mu}{\sigma}$$

using dataset-specific mean $\mu$ and standard deviation $\sigma$.

## 3.2 MODEL ARCHITECTURE

GRAFTa consists of three sequential stages that transform audio spectrograms into class predictions through graph-based reasoning (Figure 1):

1. **CNN Encoder → Node Features** A pretrained CNN14 backbone (with frozen weights except for the final layer) extracts per-frame feature embeddings from the input spectrogram. The encoder processes $S \in \mathbb{R}^{3 \times 128 \times T}$ and outputs per-frame embeddings:

   $$\mathcal{F} = \text{CNN14}(S) \in \mathbb{R}^{N \times D}$$

   where $N = 33$ is the number of temporal nodes (frames) after five $2 \times 2$ pooling operations and one $1 \times 1$ pool operation in the CNN, and $D = 256$ is the feature dimension per node extracted by CNN14.

   To adapt the pre-trained CNN14 model, the single-channel spectrogram is duplicated across all three input channels to create a 3-channel tensor, allowing the frozen first layer weights to be used directly.

2. **Graph Attention Layer** We construct a sparse adjacency matrix by computing cosine similarities between all node pairs and retaining only the top-$k = 5$ neighbors per node (including self-loops), where $\mathcal{N}(i)$ denotes the neighborhood of node $i$. A multi-head Graph Attention Network then updates node features:

   $$h'_i = \|_{m=1}^{H} \sum_{j \in \mathcal{N}(i)} \alpha_{ij}^{(m)} W^{(m)} f_j$$

   where $\|$ denotes concatenation, $H = 8$ is the number of attention heads, $f_j$ are the input node features from $\mathcal{F}$, and $W^{(m)} \in \mathbb{R}^{d' \times D}$ are learnable projection matrices mapping from input dimension $D = 256$ to per-head dimension $d' = 64$. Attention weights are computed as:

   $$\alpha_{ij}^{(m)} = \frac{\exp(\text{LeakyReLU}(a^{(m)T}[W^{(m)} f_i \| W^{(m)} f_j]))}{\sum_{k \in \mathcal{N}(i)} \exp(\text{LeakyReLU}(a^{(m)T}[W^{(m)} f_i \| W^{(m)} f_k]))}$$

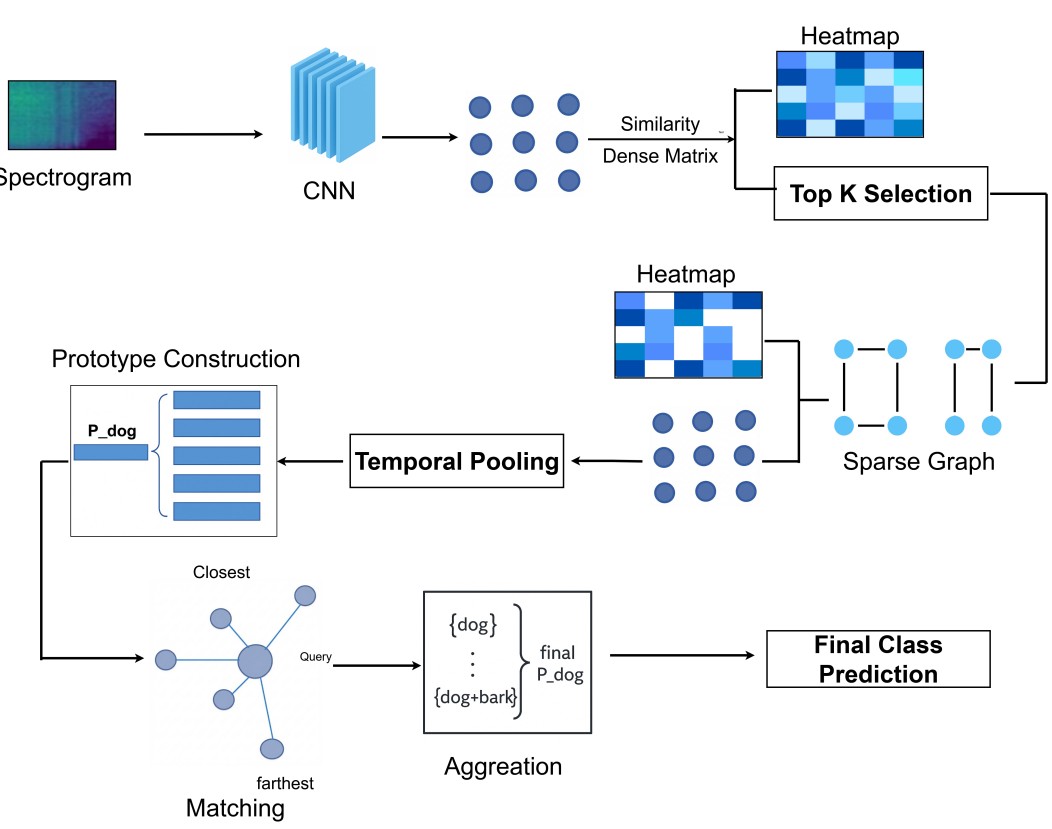

Figure 1: Overall architecture of the proposed system, including spectrogram input, CNN backbone, GAT block, and the prototypical network.

where $a^{(m)} \in \mathbb{R}^{2d'}$ is a learnable attention vector for head $m$. We use LeakyReLU with slope 0.3, and apply residual connections followed by LayerNorm to the concatenated multi-head output $h'_i \in \mathbb{R}^{H \cdot d'} = \mathbb{R}^{256}$.

3. **Prototype Matching** Our prototype-based classifier maintains two types of prototypes:

- *Single-class prototypes* $\{p_c\}_{c=1}^{C}$, where $C$ is the number of classes in the episode and each $p_c \in \mathbb{R}^{256}$ represents an individual class.
- *Compositional subset prototypes* representing frequently co-occurring label combinations mined from the support set.

During few-shot episodes, we compute prototypes for each class or subset $s$ by averaging support set node features, where $|S|$ denotes the number of support examples for that class/subset:

$$p_s = \frac{1}{|S|} \sum_{x_i \in S} \left( \frac{1}{N} \sum_{t=1}^{N} h'_{i,t} \right)$$

where $h'_{i,t} \in \mathbb{R}^{512}$ is the GAT output for node $t$ of sample $i$, and the inner sum performs temporal pooling over the $N = 33$ nodes, yielding $p_s \in \mathbb{R}^{512}$.

For query classification, node features $h'_q$ are temporally pooled to $\bar{h}'_q \in \mathbb{R}^{512}$, and class probabilities are computed using cosine similarity to prototypes:

$$\ell_s(q) = \frac{\bar{h}'_q \cdot p_s}{||\bar{h}'_q|| \cdot ||p_s||}, \quad \hat{y}_s = \sigma\big(\ell_s(q)/\tau\big)$$

where $\tau = 0.4$ is a temperature parameter and $\sigma$ is the sigmoid function.

**Subset-to-Class Aggregation**  Class scores are obtained by aggregating over subsets that contain the class:

$$\hat{y}_c = \frac{\sum_{s:c\in s} w(s)\, \hat{y}_s}{\sum_{s:c\in s} w(s)}, \quad w(s) = \frac{1}{|s|}.$$

Only subsets present in the current episode (appearing in either support or query sets) are included in the prototype computation, effectively masking out irrelevant label combinations. This episode-specific masking strategy focuses computational resources on label patterns relevant to the current few-shot task.

**Loss**  We optimize Focal loss Lin et al. (2017) on the aggregated class predictions.

**Design Validation**  We validated key architectural choices through targeted ablation studies (Section 4). Removing compositional subset prototypes and relying solely on single-class prototypes led to substantial performance degradation, confirming the importance of explicitly modeling label co-occurrence patterns.

However, we found that aggressive pruning of subset prototypes is essential for computational feasibility. Retaining all discovered subsets caused the training process to become memory-bound and unstable. Our approach of keeping only 256 of the possible subsets provides an effective balance between model expressiveness and computational efficiency, consistent with findings in recent work on compositional few-shot learning. (see Supplementary Material Section 1.2 for a detailed K-value sweep).

## 4 ABLATION STUDIES

In this section, we conduct comprehensive ablation studies to validate the key design choices in GRAFTa and analyze the computational characteristics of our approach.

### 4.1 COMPOSITIONAL VS. SINGLETON PROTOTYPES VS. LABELGRAPH CRF

To validate the necessity of compositional subset prototypes, we compare GRAFTa's default configuration against a variant that uses only singleton (per-class) prototypes and a variant that uses LabelGraph CRF. Table 1 shows the results on FSD50K. We discuss our Label Graph CRF method in more detail in the Supplementary Material (Section 1.1).

Table 1: Comparison of GRAFTa with and without subset prototypes on FSD50K.

| Prototype Type | mAP | Macro-F1 |
| --- | --- | --- |
| Subset (default) | 0.2296 | 0.0451 |
| Singleton only | 0.0035 | 0.0005 |
| LabelGraph CRF | 0.1499 | – |

The sharp performance drop with singleton-only prototypes highlights how vital it is to model label co-occurrences. In few-shot multi-label scenarios, single-class prototypes alone lack the context needed to disambiguate overlapping sounds—for instance, an "engine" prototype cannot reliably separate "engine + siren" from "engine + speech." Forcing the model to represent complex co-occurrences with such limited, ambiguous prototypes leads to overprediction and poorly calibrated outputs. We discuss our Label Graph CRF method in more detail in our supplementary material

Our subset-based approach mitigates this by explicitly modeling common label combinations as compositional prototypes, allowing more expressive and context-sensitive matching. This confirms the importance of modeling class co-occurrence in few-shot multi-label audio tagging.

### 4.2 EFFECT OF SHOT COUNT

We evaluate GRAFTa's performance under different shot counts (1-shot, 3-shot, 5-shot, and 10-shot) on FSD50K using 200 few-shot episodes per setting. Table 2 presents the relationship between support set size and prediction behavior.

Table 2: Effect of shot count on prediction behavior and mAP on FSD50K (averaged over 200 episodes).

| Shot Count | True Positives | Predicted Positives | mAP |
|---|---|---|---|
| 1-shot | 2 | 2 | 1.0000 |
| 3-shot | 48 | 125 | 0.3794 |
| 5-shot | 152 | 842 | 0.2296 |
| 10-shot | 312 | 9302 | 0.1250 |

While 1-shot yields artificially high mAP due to extremely low positive label counts, it is not reliable for practical evaluation. As the number of support examples increases, we observe a steep increase in predicted positives relative to true positives, reflecting growing prototype overlap and class confusion. This behavior highlights a key challenge in few-shot multi-label learning: balancing prototype expressiveness with prediction precision as the support set grows.

**Additional Observations**    Our analysis reveals several key insights about GRAFTa's behavior:

- Large prototype sets lead to overlapping cosine similarities, which can cause prediction ambiguity
- Query-time overprediction grows with episode shot count, suggesting the need for better calibration mechanisms
- Singleton-only models are more conservative in their predictions but achieve significantly lower precision

These findings inform future work on improving prototype distinctiveness and prediction calibration in few-shot multi-label scenarios. For detailed calibration analysis including Expected Calibration Error (ECE) across both datasets, see Supplementary Material Section 2.

## 5    EXPERIMENTS

### 5.1    DATASETS

We evaluate **GRAFTa** on two widely used audio tagging benchmarks:

- **FSD50K** Fonseca et al. (2021): A 200-class dataset containing 37,134 training, 4,170 validation, and 10,231 test clips. The clips range from 0.3 to 30 seconds in duration and often include multiple overlapping sound events.
- **AudioSet–Balanced** (Gemmeke et al., 2017): A curated 527-label subset of AudioSet consisting of 20,550 training clips and 1,887 evaluation clips. Each 10-second clip may contain several co-occurring acoustic events.

For both datasets, we follow the original data splits provided in their respective papers to ensure comparability with prior work and to evaluate the model under realistic, real-world conditions.

Few-shot multi-label classification poses unique challenges because single-class prototypes alone fail to capture the interactions between overlapping events. For example, an "engine" prototype cannot reliably distinguish an "engine + siren" combination from "engine + speech." When complex co-occurrences must be represented by such limited and ambiguous prototypes, the model tends to overpredict labels and produce poorly calibrated outputs. GRAFTa directly addresses this limitation by incorporating relational structure into the prototype space.

### 5.2    IMPLEMENTATION DETAILS

**Feature Extraction and Augmentation**    All clips are converted to mono, resampled to 22.05 kHz, and padded/tiled to 10 s. We extract 128-bin log-Mel spectrograms (FFT 2048, hop 209), replicate them to 3 channels, and normalize using per-dataset mean/std We apply Mixup (Beta distribution, $\alpha = 0.5$) and SpecAugment with 15% frequency masking and 20% time masking during training.

**Model Architecture**    We use a pretrained CNN14 encoder from PANNs (Kong et al., 2020), frozen except for the final layer, to extract spectrogram features. to extract frame-level features from the spectrogram. Features then feed into a single GAT layer (8 heads, hidden dim 128, top-k=5 neighbors, LeakyReLU $\alpha = 0.3$). For metric learning we maintain both singleton and compositional prototypes: subsets (up to size 3) are mined on-the-fly from support labels and we keep the top K=256 most frequent ones. The GAT and all prototype parameters are trained from scratch.

**Training Protocol**    We adopt a meta-learning framework and train directly on few-shot episodes. Each episode consists of 5 classes, 5 support examples per class, and 5 query examples per class (i.e., 5-way 5-shot 5-query). We train for 10,000 episodes using the Adam optimizer (learning rate $3\mathrm{e}{-}4$, weight decay $1\mathrm{e}{-}4$), with a 0.5 learning rate decay every 3,000 episodes. We use a batch size of 1 episode per step. All training is conducted on a single GPU and completes in under 12 hours.

**Few-Shot Evaluation:**    We focus on **few-shot episodic evaluation**, where performance is measured across randomly sampled test episodes. We sample 200 episodes from the test split (5-way 5-shot 5-query) and compute average mean Average Precision (mAP) and macro-F1 across all query examples. Query labels may include multiple active classes, making this a multi-label few-shot setting.

## 5.3    Baselines and Metrics

Since prior work hasn't benchmarked few-shot multi-label audio tagging systematically, direct comparisons with existing models require major tweaks. Fully supervised models such as PANNs Kong et al. (2020) and AST Gong et al. (2021) demand large-scale end-to-end training and aren't built for episodic few-shot setups. Likewise, graph-based methods such as SpecGraph Lu et al. (2022) assume full supervision and single-label settings, making direct application to few-shot multi-label tasks infeasible without substantial modification.

Accordingly, we treat GRAFTa's results as strong initial baselines for this new task setup. To evaluate performance, we report:

- **Mean Average Precision (mAP)** — averaged over all classes and episodes.

All metrics are reported across 200 randomly sampled episodes (5-way, 5-shot, 5-query) from the test split. These settings align with standard few-shot evaluation protocols in the image and audio domains Snell et al. (2017); Wang et al. (2020). All metrics are reported across 200 randomly sampled episodes (5-way, 5-shot, 5-query) from the test split. Detailed per-class performance analysis and label cardinality breakdowns are provided in Supplementary Material Section 3.

## 6    Results

### 6.1    Few-Shot Multi-Label Evaluation

We evaluate GRAFTa in a 5-way 5-shot setting on both FSD50K and AudioSet-Balanced. Table 3 reports mean Average Precision (mAP) across 200 test episodes.

| Model | FSD50K | Audioset-Balanced | Per Node Dimension |
|---|---|---|---|
| GRAFTa | $0.2271 \pm 0.0575$ | $0.2302 \pm 0.0557$ | 256 |
| CNN + ProtoNet | $0.2056 \pm 0.0441$ | $0.1936 \pm 0.0466$ | 256 |

Table 3: Few-shot (5-way, 5-shot) multi-label mAP performance.

The use of a compositional subset prototype mechanism enables the model to generalize beyond individual labels and capture meaningful co-occurrence patterns between acoustic events. The use of a compositional subset prototype mechanism enables the model to generalize beyond individual labels and capture meaningful co-occurrence patterns between acoustic events. Comprehensive per-class AUPRC analysis is available in Supplementary Material Section 3.

## 6.2 COMPARISON AND BENCHMARKING

As few-shot multi-label tagging on FSD50K and AudioSet-Balanced has not been previously bench-marked, we provide these results as the first reference point for this task. Due to architectural and training incompatibilities, standard supervised models such as PANNs Kong et al. (2020) and AST Gong et al. (2021) are not directly comparable in this setting without extensive modification or retraining, and are therefore omitted from this evaluation.

# 7 LIMITATIONS

While GRAFTa achieves strong results in both few-shot and single-pass regimes, several limitations remain:

- **Limited Few-Shot Benchmarks** Due to a lack of existing few-shot *multi-label* benchmarks on FSD50K and AudioSet-Balanced, especially with mAP and macro-F1 reporting, direct comparisons to prior work are limited. We adopt reasonable baselines and protocols, but standardization remains an open need in the community.
- **Generalization Scope** GRAFTa is evaluated on two curated datasets. Its robustness to out-of-domain sounds, longer-tail class distributions, or real-world noise remains to be tested.

# 8 CONCLUSION

In this work, we introduced GRAFTa, a lightweight, graph-based framework for few-shot multi-label audio tagging. Unlike traditional tagging pipelines, GRAFTa models spectrograms as graphs and leverages compositional prototypes to capture co-occurring label structures with minimal super-vision. Our results on FSD50K and AudioSet-Balanced demonstrate that, even under challenging low-shot conditions, structured inductive biases can enable meaningful generalization.

While GRAFTa does not aim to outperform fully supervised models, it opens a new direction in few-shot acoustic learning especially for scenarios where data is scarce and label overlap is common. There is still much to explore: better training strategies, faster inference, richer ablations, and more robust evaluation protocols. We view this work as a first step in bridging the gap between meta-learning and multi-label audio understanding, and we hope it encourages further progress in this space.

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
