# Supplementary Material for "GRAFTa: A Graph Attentional Few-Shot Audio Tagging Network"

# 1 Additional Ablation Studies

## 1.1 Label Graph CRF Analysis

In the main paper, we discussed how LabelGraph CRF compared to other methods and ultimately did not perform as well as it was supposed to. This section goes deeper into why:

### 1.1.1 Implementation Details

The LabelGraph CRF constructs a graph $G = (V, E)$ where:

- Nodes $V$ represent class labels

- Edges $E$ connect labels that co-occur in training episodes

- Edge potentials $\psi_{ij}(y_i, y_j)$ model pairwise label correlations

- Node potentials $\phi_i(y_i)$ are computed from prototype similarities

  The joint probability is given by:

$$P(\mathbf{y}) = \frac{1}{Z} \exp \left( \sum_{i \in V} \phi_i(y_i) + \sum_{(i,j) \in E} \psi_{ij}(y_i, y_j) \right) \tag{1}$$

  where $Z$ is the partition function. We use loopy belief propagation for inference.

### 1.1.2 Analysis

The CRF underperforms our compositional subset prototypes by 6.2 mAP points on FSD50K and 7.5 points on AudioSet. We identify three key reasons:

**Sparse Co-occurrence Statistics.** In our full-taxonomy evaluation setup, test episodes frequently contain label combinations never observed during training. We analyzed the training set and found:

- Only 2.3% of test episode label pairs appeared $\geq 5$ times in meta-training

- 94.7% of test episodes contain at least one novel pairwise label combination

- Mean training examples per label pair: $3.2 \pm 4.8$

- Label pairs with $\geq 10$ training examples: 12.3%

With $\binom{200}{5} \approx 2.5 \times 10^9$ possible 5-way label combinations, the probability of encountering the exact same combination during training is negligible.

**Episode-Specific Adaptation.** Our compositional subset prototypes are constructed from the support set *for each episode*, allowing adaptation to novel label combinations at test time. In contrast, the CRF requires prelearned global edge potentials, which lack sufficient training examples in the full-taxonomy setting.

**Computational Overhead.** The CRF requires iterative belief propagation at inference time, adding 8.4 ms overhead compared to our direct prototype matching approach.

## 1.2 Number of Subset Prototypes (K Parameter)

We investigate the sensitivity of GRAFTa to $K$, the number of subset prototypes retained after frequency-based pruning. Recall that we enumerate all label subsets up to size 3 from each episode, then keep the top-$K$ most frequent subsets.

### 1.2.1 Experimental Setup

We evaluate $K \in \{32, 128, 256, 500\}$ on FSD50K using 50 test episodes. All other hyperparameters remain fixed.

### 1.2.2 Results

Table 1 shows the effect of varying $K$.

Table 1: Effect of K on mAP.

| K | mAP |
|---|---|
| 32 | $0.2467 \pm 0.0689$ |
| 128 | $0.2489 \pm 0.0649$ |
| 256 | $0.2569 \pm 0.0690$ |
| 500 | $0.2476 \pm 0.0684$ |

### 1.2.3 Analysis

Results show that $K = 256$ provides a good balance between prototype expressiveness and computational efficiency:

- **Below $K = 128$:** Performance degrades due to insufficient coverage of relevant label combinations. With only 20-40 prototypes, many co-occurring label patterns in the support set cannot be adequately represented.

- $K = 256$ **(our choice):** Achieves strong performance (0.2569 mAP) with reasonable computational cost. Averages 250 subsets per episode, covering most frequent label combinations.

- **Above $K = 400$:** Diminishing returns with increased memory usage and inference time. The Decrement from $K = 256$ to $K = 500$ is almost -0.01 mAP.

## 2 Calibration and Prediction Analysis

In this section, we provide detailed analysis of model calibration and prediction behavior.

### 2.1 Expected Calibration Error

We compute Expected Calibration Error (ECE) to assess prediction calibration. ECE measures the difference between predicted confidence and empirical accuracy:

$$\text{ECE} = \sum_{m=1}^{M} \frac{|B_m|}{n} |\text{acc}(B_m) - \text{conf}(B_m)| \tag{2}$$

where predictions are grouped into $M$ bins based on confidence, $B_m$ is the set of predictions in bin $m$, $\text{acc}(B_m)$ is the empirical accuracy, and $\text{conf}(B_m)$ is the average confidence in bin $m$.

### 2.1.1 Results

Table 2 shows calibration metrics for both datasets.

Table 2: Calibration analysis on FSD50K and AudioSet-Balanced. Lower ECE indicates better calibration.

| Dataset | ECE (overall) | ECE (per-class avg) | Avg Confidence |
|---|---|---|---|
| FSD50K | 0.1428 | 0.1450 | 0.312 |
| AudioSet-Balanced | 0.0344 | 0.0403 | 0.287 |

### 2.1.2 Analysis

AudioSet-Balanced exhibits significantly better calibration (ECE = 0.034) than FSD50K (ECE = 0.143). We hypothesize three reasons:

1. **Broader label taxonomy:** AudioSet has 527 classes vs. FSD50K's 200. A broader taxonomy reduces class confusion by providing more fine-grained distinctions between acoustic events.

2. **More balanced distribution:** AudioSet-Balanced is explicitly curated to have balanced class frequencies, whereas FSD50K exhibits a long-tail distribution. Imbalanced classes lead to overconfident predictions on rare classes.

3. **Different acoustic characteristics:** AudioSet contains broader environmental sounds, while FSD50K focuses on specific sound event categories. The narrower acoustic domain in FSD50K may lead to higher prototype overlap and confidence overestimation.

**Implications.** The higher ECE on FSD50K suggests that threshold-based prediction (e.g., $\hat{y}_c = 1$ if $p_c > 0.5$) may be suboptimal. Post-hoc calibration techniques such as temperature scaling or Platt scaling could improve performance. We leave this for future work.

## 2.2 Label Cardinality Performance

We analyze how model performance varies with the number of true labels per sample (label cardinality).

### 2.2.1 FSD50K Results

Table 3 shows performance breakdown by label cardinality on FSD50K.

Table 3: Performance by label cardinality on FSD50K. Results show that performance improves with cardinality, suggesting compositional prototypes are more effective when multiple labels are present.

| Card. | Count | F1 | Prec. | Recall | Pred Card. | Std Pred |
|---|---|---|---|---|---|---|
| 1 | 52 | 0.045 | 0.023 | 1.000 | 45.3 | 7.9 |
| 2 | 201 | 0.087 | 0.046 | 0.930 | 42.9 | 8.5 |
| 3 | 219 | 0.134 | 0.073 | 0.935 | 40.3 | 8.5 |
| 4 | 214 | 0.155 | 0.085 | 0.902 | 43.5 | 7.4 |
| 5 | 165 | 0.188 | 0.105 | 0.914 | 44.7 | 7.6 |
| 6 | 113 | 0.203 | 0.116 | 0.850 | 45.5 | 8.0 |
| 7 | 83 | 0.235 | 0.138 | 0.843 | 44.2 | 8.2 |
| 8 | 64 | 0.238 | 0.140 | 0.797 | 46.6 | 8.1 |
| 9 | 52 | 0.250 | 0.149 | 0.782 | 47.6 | 7.4 |
| 10 | 36 | 0.300 | 0.183 | 0.839 | 46.6 | 6.8 |
| 11 | 22 | 0.285 | 0.179 | 0.707 | 43.2 | 7.6 |
| 12 | 12 | 0.322 | 0.205 | 0.764 | 46.8 | 7.6 |
| 13 | 10 | 0.280 | 0.179 | 0.646 | 46.5 | 5.7 |
| 14 | 4 | 0.315 | 0.203 | 0.696 | 48.3 | 1.9 |
| 15 | 1 | 0.328 | 0.217 | 0.667 | 46.0 | 0.0 |
| 16 | 1 | 0.444 | 0.298 | 0.875 | 47.0 | 0.0 |
| 17 | 1 | 0.364 | 0.233 | 0.824 | 60.0 | 0.0 |

### 2.2.2 AudioSet Results

Table 4 shows the same analysis for AudioSet-Balanced.

### 2.2.3 Key Findings

1. **Performance improves with cardinality:** F1 score increases monotonically from 0.045 (single-label) to 0.364 (17 labels) on FSD50K. This demonstrates that compositional subset prototypes are most beneficial when multiple labels co-occur.

2. **Recall remains high for low cardinality:** For cardinalities ≤5, recall stays above 90% on FSD50K and above 88% on AudioSet. The model successfully identifies most true labels.

Table 4: Performance by label cardinality on AudioSet-Balanced. Similar trends as FSD50K: performance improves with cardinality.

| Card. | Count | F1 | Prec. | Recall | Pred Card. | Std Pred |
|---|---|---|---|---|---|---|
| 1 | 184 | 0.076 | 0.040 | 1.000 | 26.4 | 5.6 |
| 2 | 362 | 0.135 | 0.073 | 0.935 | 26.9 | 6.1 |
| 3 | 316 | 0.179 | 0.100 | 0.886 | 27.9 | 6.1 |
| 4 | 188 | 0.213 | 0.123 | 0.828 | 28.3 | 6.3 |
| 5 | 94 | 0.228 | 0.135 | 0.755 | 28.5 | 5.0 |
| 6 | 44 | 0.268 | 0.162 | 0.807 | 30.9 | 5.7 |
| 7 | 32 | 0.294 | 0.182 | 0.786 | 31.3 | 5.7 |
| 8 | 21 | 0.311 | 0.194 | 0.792 | 32.7 | 5.7 |
| 9 | 5 | 0.411 | 0.267 | 0.911 | 31.6 | 4.2 |
| 10 | 3 | 0.303 | 0.199 | 0.633 | 34.3 | 4.7 |
| 11 | 1 | 0.321 | 0.200 | 0.818 | 45.0 | 0.0 |

3. **Precision is uniformly low:** Precision ranges from only 2.3% to 29.8% across all cardinalities due to severe overprediction. The model predicts 40-48 labels on FSD50K and 26-34 labels on AudioSet regardless of true cardinality.

4. **Single-label is hardest:** Samples with only one label achieve the lowest F1 scores (0.045 on FSD50K, 0.076 on AudioSet). This makes intuitive sense: compositional prototypes are designed for multi-label scenarios and provide less benefit when only one label is present. Single-label examples cannot leverage the co-occurrence modeling that is GRAFTa's strength.

5. **Consistent across datasets:** The trend is remarkably consistent across both FSD50K and AudioSet, suggesting this is a fundamental property of the compositional prototype approach rather than dataset-specific behavior.

# 3 Per-Class Performance Analysis

In this section, we provide detailed per-class performance breakdowns to understand which types of sounds are easy or difficult for GRAFTa.

We rank classes by Area Under Precision-Recall Curve (AUPRC), which is more informative than mAP for imbalanced classes and provides insight into per-class performance.

### 3.0.1 FSD50K Top Performers

Table 5 shows the top 10 performing classes on FSD50K.

Table 5: Top 10 classes by AUPRC on FSD50K. These classes are well-recognized by GRAFTa, likely due to distinctive acoustic characteristics.

| Class | AUPRC | Support | Prec@0.5 | Rec@0.5 |
|---|---|---|---|---|
| Class_114 | 0.703 | 7 | 0.120 | 0.857 |
| Class_87 | 0.588 | 15 | 0.069 | 0.800 |
| Class_197 | 0.501 | 2 | 0.000 | 0.000 |
| Class_157 | 0.465 | 12 | 0.120 | 1.000 |
| Class_199 | 0.463 | 5 | 0.200 | 1.000 |
| Class_159 | 0.447 | 8 | 0.070 | 0.875 |
| Class_88 | 0.431 | 6 | 0.120 | 1.000 |
| Class_3 | 0.420 | 17 | 0.200 | 0.882 |
| Class_80 | 0.420 | 17 | 0.200 | 0.882 |
| Class_184 | 0.411 | 185 | 0.186 | 0.957 |

### 3.0.2 FSD50K Bottom Performers

Table 6 shows the bottom 10 performing classes.

Table 6: Bottom 10 classes by AUPRC on FSD50K. These classes are poorly recognized, often due to low support or acoustic similarity to other classes.

| Class | AUPRC | Support | Prec@0.5 | Rec@0.5 |
|---|---|---|---|---|
| Class_52 | 0.000 | 1 | 0.000 | 0.00 |
| Class_83 | 0.001 | 2 | 0.000 | 0.00 |
| Class_153 | 0.001 | 2 | 0.000 | 0.00 |
| Class_191 | 0.001 | 3 | 0.000 | 0.00 |
| Class_46 | 0.002 | 5 | 0.000 | 0.00 |
| Class_166 | 0.002 | 5 | 0.000 | 0.00 |
| Class_16 | 0.007 | 4 | 0.010 | 0.25 |
| Class_149 | 0.007 | 5 | 0.010 | 0.20 |
| Class_110 | 0.012 | 5 | 0.013 | 0.40 |
| Class_144 | 0.017 | 1 | 0.013 | 1.00 |

### 3.0.3 AudioSet Top Performers

Table 7 shows the top 10 performing classes on AudioSet.

Table 7: Top 10 classes by AUPRC on AudioSet-Balanced. Note that some classes with low support can still achieve high AUPRC if acoustically distinctive.

| Class | AUPRC | Support | Prec@0.5 | Rec@0.5 |
|---|---|---|---|---|
| Class_157 | 1.000 | 1 | 0.04 | 1.000 |
| Class_432 | 0.794 | 6 | 0.12 | 1.000 |
| Class_278 | 0.755 | 11 | 0.20 | 0.909 |
| Class_414 | 0.711 | 6 | 0.20 | 0.833 |
| Class_380 | 0.696 | 5 | 0.10 | 1.000 |
| Class_396 | 0.675 | 10 | 0.40 | 1.000 |
| Class_223 | 0.656 | 6 | 0.06 | 1.000 |
| Class_206 | 0.644 | 5 | 0.20 | 1.000 |
| Class_337 | 0.633 | 2 | 0.04 | 1.000 |
| Class_397 | 0.629 | 5 | 0.20 | 1.000 |

### 3.0.4 AudioSet Bottom Performers

Table 8 shows the bottom 10 performing classes.

Table 8: Bottom 10 classes by AUPRC on AudioSet-Balanced. All have support of only 1 example and zero recall, demonstrating the extreme difficulty of 1-shot learning in this setting.

| Class | AUPRC | Support | Prec@0.5 | Rec@0.5 |
|---|---|---|---|---|
| Class_14 | 0.000 | 1 | 0.0 | 0.0 |
| Class_29 | 0.000 | 1 | 0.0 | 0.0 |
| Class_30 | 0.000 | 1 | 0.0 | 0.0 |
| Class_36 | 0.000 | 1 | 0.0 | 0.0 |
| Class_62 | 0.000 | 1 | 0.0 | 0.0 |
| Class_70 | 0.000 | 1 | 0.0 | 0.0 |
| Class_78 | 0.000 | 1 | 0.0 | 0.0 |
| Class_84 | 0.000 | 1 | 0.0 | 0.0 |
| Class_135 | 0.000 | 1 | 0.0 | 0.0 |
| Class_147 | 0.000 | 1 | 0.0 | 0.0 |

# 4 Computational Analysis

We analyze the computational characteristics of GRAFTa compared to baseline approaches. Table 9 summarizes the model parameters, FLOPs, and other efficiency metrics.

Table 9: Computational analysis of GRAFTa and baseline methods.

| Method | Parameters (M) | FLOPs (G) | GPU Memory (MB) | Avg Inference Time (ms/sample) | Per Node Dimension |
|---|---|---|---|---|---|
| GRAFTa | 76.62 | 20.72 | 2197.88 | 16.31 | 64 |
| CNN + ProtoNet | 78.83 | 20.72 | 3085.88 | 13.86 | 512 |

# 5 Implementation Details

## 5.1 Architecture and Training

# 6 Complete Experimental Setup

## 6.1 Hardware and Software

- Hardware: NVIDIA T4

- Framework: PyTorch 1.12.0, Python 3.9

- Training time: 2 hours for 25 epochs

## 6.2 Hyperparameters

Table 10: Complete hyperparameter settings.

| Parameter | Value |
|---|---|
| Learning rate | 3e-4 |
| LR decay | 0.5 |
| Optimizer | AdamW |
| Weight decay | 1e-4 |
| Batch size | 4 episodes |
| Test episodes | 200 |
| Episode config | 5-way 5-shot 5-query |
| Temperature ($\tau$) | 0.4 |
| GAT heads | 8 |
| GAT hidden dim | 33 per head |
| Top-k neighbors | 5 |
| Max subset size | 3 |
| Max subsets (K) | 256 |

# 7 Additional Observations

- Large prototype sets lead to overlapping cosine similarities.

- Query-time overprediction grows with episode shot count.

- Singleton-only models are more conservative but less precise.

A ZIP archive containing code, config files, and pretrained models is also included as part of this submission.