# OpenReview forum: "GRAFTa: A Graph Attentional Few-Shot Audio Tagging Network"
_ICLR.cc/2026/Conference — ICLR 2026 Conference Desk Rejected Submission_

### Official Review · Reviewer_Ko7d · 2025-10-25

**Soundness:** 2
**Presentation:** 2
**Contribution:** 2
**Rating:** 2
**Confidence:** 3

**Summary:**

The paper proposes GRAFTa, a few-shot multi-label audio tagging framework that (i) builds a graph over spectrogram time–frequency regions, processes it with a GAT block, and (ii) performs prototype matching not only for single labels but also for compositional subset prototypes (frequently co-occurring label sets mined from the episode’s support set, up to size 3). The model uses a CNN14/PANNs backbone for per-frame features, constructs a sparse k-NN graph (top-k=5) with multi-head GAT (H=8), and aggregates node embeddings to compare against class/subset prototypes via cosine similarity with temperature. Class scores are aggregated from subset scores with size-weighted averaging and trained using Focal Loss. Evaluation follows episodic 5-way 5-shot protocols on FSD50K and AudioSet-Balanced, reporting mAP; the authors also present ablations on singleton vs. subset prototypes and shot count, and a small compute analysis.

**Strengths:**

* Originality: Introduces compositional subset prototypes in a multi-label few-shot setting; integrates GAT over spectrogram nodes to address non-local dependencies.
* Quality (technical): The pipeline looks coherent; subset-to-class aggregation and temperature-scaled cosine scoring are well-specified; focal loss is a reasonable choice for imbalance.
* Significance (potential): Opens a few-shot multi-label benchmark track for audio; subset prototypes could inspire follow-ups on label-dependency modeling and calibration.

**Weaknesses:**

1. Method section is not perfectly written. Many notation used are not explained or defined. For example, what $F$ in line 108. This makes readabilty poor.   In general this reviewer feel that there is a large room for presentation improvement.
2. Lack of statistical rigor: No CIs, standard deviations, or multi-seed runs; difficult to judge stability or significance.
3. Evaluation design & metrics: Over-reliance on averaged mAP; limited per-class analysis, no label-cardinality breakdowns, and calibration metrics are missing (e.g., AUPRC per class, macro vs. micro, expected calibration error). Shot-count table suggests metric artifacts that are not investigated.
4. Baselines limited: No adapted strong audio encoders (e.g., AST/HTS-AT/CLAP-like features with ProtoNet or matching networks) under identical episodic protocols; graph-free but label-dependency models (e.g., sigmoid-CRF, classifier chains, bipartite attention over labels) are not compared.

**Questions:**

1. You state pruning “first 500 subsets” after sorting, but later “retain top K=80 most frequent subsets.” Which is used in the reported results? How sensitive is mAP to K? Please provide a sweep (e.g., 32/64/128/256/512).
2. Did you tune per-class thresholds or use any calibration (e.g., Platt/temperature scaling) across episodes? Over-prediction grows with shot count; could a class-wise threshold or subset-wise sparsity penalty address this?
3. Have you compared subset prototypes to label-graph CRF or attention over the label set (without exponential subset growth)? This could control the combinatorial explosion while retaining dependency structure.

---

> ### Author Response · Authors · 2025-11-21
> **Response to concerns on Notation, Rigor and Evaluation**
>
> Thank you for your feedback. We will try to answer all of it.
> 1. We apologize for the confusion about the K value. K=80 was used in our originally reported results. However, now we use K=256. We have now conducted the sweep the reviewer requested, trained on 10 epochs, and evaluated on 50 episodes:
>
> | K | mAP |
> |----------|----------|
> | 32    | 0.2467 +- 0.0689  |
> | 128    | 0.2489 +- 0.0649  |
> | 256    | 0.2569 +- 0.0690 |
> | 500    | 0.2476 +- 0.0684  |
>
> Performance improves up to K=256 and decreases at K=500, suggesting diminishing returns from excessive subset prototypes. Full results will be in the supplementary materials.
>
> 2. We ran all experiments again and now reports standard deviations as well. Also, updated results are included above and will be in the revised PDF
>
> 3. We have revised the methodology section to explicitly define all notation upon first use, including the subscript in line 108 that you found. Thank you for noticing
>
> 4. When it comes to alternative methods,
> We have implemented a label-graph CRF baseline as you suggested. Under our episodic protocol, it achieves approximately 0.14 mAP on FSD50K We were not able to get standard deviations for this result; however, since it's substantially below GRAFTa's performance, we find that it's not necessary. We believe this is because CRF-based approaches require more training data to learn stable label transition parameters, whereas our subset prototype approach can be computed directly from the support set without learned parameters. Another reason was because of the nature of multi-label classification. 5 shots does not mean 5 support labels, usually due to clips having multiple labels, it goes from 27-31. Hence, it adds another layer that makes Label Graph CRF not as good as subset prototypes. We will add this comparison to our results and go further in the supplementary material. Thank you for your input
>
> 5. We are computing additional metrics (AUPRC, macro/micro breakdown, per-class analysis by class frequency) and will include these in the revised manuscript and supplementary materials.
>
> 6. Regarding the Shot Count Table (Table 2) Our temperature parameter (τ=0.4) partially mitigates this, but we acknowledge that better calibration mechanisms are needed. We will add this comparison to our supplementary material in our revised PDF
>
> Thank you so much for your constructive criticism and review once again. We are currently working on our revised paper and incorporating all the changes discussed above, along with additional feedback from the other reviewers. We’ll be uploading the revised PDF and supplementary materials within the rebuttal window. We wanted to respond quickly to acknowledge the reviewers’ concerns and share our new experimental results. We’ll post a follow-up comment once the full revision is ready.

---

> ### Author Response · Authors · 2025-11-25
>
> Thank you for taking the time to review our revised approach and for your thoughtful consideration. Yes, we will also post a public comment highlighting every change and addition made in the revised paper.

---

### Official Review · Reviewer_x997 · 2025-10-29

**Soundness:** 3
**Presentation:** 3
**Contribution:** 2
**Rating:** 4
**Confidence:** 4

**Summary:**

This paper proposes a model for few-shot audio tagging based on a graph attention network. Experiments are carried out on the FSD50k and AudioSet-balanced datasets, and comparisons are made with a CNN baseline.

**Strengths:**

* The paper is overall clear, well written and well structured.
* The motivation behind the proposed method is timely, and the overall methodology including data preparation and model architecture is technically correct.

**Weaknesses:**

* There are relevant works on using graph neural networks for audio classification which have not been cited nor compared against. Similarly there are works on few-shot audio classification/tagging which have also not been cited nor compared against.
* Citations in the main paper are not appropriately presented and formatted - please take care of using \cite vs. \citep.
* Before presenting section 4, a section on the data used would have been useful to have (which is subsection 5.1).

**Questions:**

* Figure 1 is quite generic, does not include axes labels, does not include any information on the sounds included in the spectrograms, and does not contribute to the main contribution of this work. It can be removed entirely without any loss of information, or it could be revised perhaps to indicate how GNNs could contribute for audio tagging.
* Citations in the main paper are not appropriately presented and formatted - please take care of using \cite vs. \citep.
* From section 2 there is one relevant paper being cited (Lu et al 2022) however there is no discussion on how is the proposed model different from that previous one (despite the lack of a few-shot setting which is not about the model itself). There are also other relevant works on using GNNs for audio tagging that would need to be cited and compared against, e.g. [i].
* Section 4 which is about ablation studies mentions some of the specific datasets used in this work, however the datasets have not been presented as of yet - please consider cross-referencing with section 5.1 on data.
* Section 5.4 mentions that few-shot multilabel audio tagging has not been systematically benchmarked. From a search I was able to find at least two relevant works [ii], [iii], [iv]. I would suggest that any relevant works on multi-label few-shot learning for audio would need to be cited and discussed, and when possible compared against the proposed model.


[i] S. Singh, C. J. Steinmetz, E. Benetos, H. Phan, and D. Stowell, “ATGNN: audio tagging graph neural network”, IEEE Signal Processing Letters, vol. 31, pp. 825-829, 2024.
[ii] K. -H. Cheng, S. -Y. Chou and Y. -H. Yang, "Multi-label Few-shot Learning for Sound Event Recognition," 2019 IEEE 21st International Workshop on Multimedia Signal Processing (MMSP), Kuala Lumpur, Malaysia, 2019.
[iii] J. Liang, H. Phan and E. Benetos, "Learning from Taxonomy: Multi-Label Few-Shot Classification for Everyday Sound Recognition," ICASSP 2024 - 2024 IEEE International Conference on Acoustics, Speech and Signal Processing (ICASSP), Seoul, Korea, Republic of, 2024, pp. 771-775.
[iv] Y. Wang, N. J. Bryan, M. Cartwright, J. Pablo Bello and J. Salamon, "Few-Shot Continual Learning for Audio Classification," ICASSP 2021 - 2021 IEEE International Conference on Acoustics, Speech and Signal Processing (ICASSP), Toronto, ON, Canada, 2021, pp. 321-325.

---

> ### Author Response · Authors · 2025-11-21
> **Reply to Reviewer feedback**
>
> Thank you for reviewing our research paper. We will try to answer your questions one by one.
>
> 1. We have reviewed all four papers:
> Cheng et al. 2019
> Liang et al. 2024
> Singh et al. 2024
> Wang et al. 2021
> We will cite and discuss these in our revised manuscript. However, we note an important distinction: previous multi-label few-shot audio work uses a base/novel class split, or they split their dataset based on labels. Our setting is fundamentally different — we use all 200 labels episodically with no base class pretraining, and we split our dataset based on data as the original papers did. This makes our problem harder and direct numerical comparison infeasible without reimplementing their methods under our protocol. We will clarify this distinction and properly position our contribution relative to these works. Our original paper did not go into much detail on how our datasets were made, and hence, we understand the confusion and have made it clearer in our revised paper. Thank you
>
> 2. We have corrected all instances of \cite vs \citep in the revised manuscript.
>
> 3. The original Figure 1 was not informative enough. We have replaced it with a visualization showing the spectrogram-to-graph conversion process, illustrating how time-frequency regions become nodes and how edges are constructed.
>
> 4. We have reorganized the paper so that dataset information (Section 5.1) is referred to earlier when ablation studies mention specific datasets. We also removed any mentions of datasets in section 4 and instead referenced section 5.1 in their place
>
> Thank you once again for reviewing our paper. We are currently working on our revised paper and incorporating all the changes discussed above, along with additional feedback from the other reviewers. We’ll be uploading the revised PDF and supplementary materials within the rebuttal window. We wanted to respond quickly to acknowledge the reviewers’ concerns and share our new experimental results. We’ll post a follow-up comment once the full revision is ready.

---

### Official Review · Reviewer_p2Af · 2025-10-31

**Soundness:** 2
**Presentation:** 2
**Contribution:** 1
**Rating:** 2
**Confidence:** 5

**Summary:**

This paper introduces GRAFTa, a Graph Attentional Few-shot Audio Tagging framework that models spectrogram regions as graph nodes to better capture both local and global acoustic dependencies. It further proposes compositional subset prototypes to enhance multi-label generalization from limited examples by representing frequent co-occurring label subsets. Experiments on FSD50K and AudioSet-Balanced demonstrate competitive performance and establish the first benchmarks for few-shot multi-label audio tagging.

**Strengths:**

Effectively models non-local and irregular spectrogram structures using graph attention.

**Weaknesses:**

1. Its performance can be highly sensitive to the quality of subset mining from limited support examples.
2. The evaluation is restricted to few-shot settings without comparison to large-scale pretrained or stronger baseline models.
3. The performance evaluation only reports mAP, and the compared methods are not representative, providing insufficient evidence of superiority.
4. The overall formatting of the paper is problematic (e.g., layout, figures, and references) and requires careful revision.
5. The number of cited works is very limited, and many references are outdated, failing to reflect recent progress in few-shot learning and audio tagging.

Overall, I believe the paper’s quality falls significantly below the standards of ICLR.

**Questions:**

Figure 2 has very poor structure and conveys little useful information about the model design or data flow, making it hard to understand or reproduce the method.

---

> ### Author Response · Authors · 2025-11-21
> **Reply to Reviewer's concerns**
>
> Thank you for reviewing our research paper. We will now try to address your concerns one by one.
>
> 1. while this is a valid concern, our design includes safeguards against poor subset mining: (1) singleton prototypes are always included regardless of co-occurance patterns existing or not, providing a fallback when subset mining is unreliable; (2) our new K sweep table (to be added in supplementary material) shows stable performance across K=32 to K=256, indicating robustness to the exact subset selection; and (3) the 5-shot setting provides 25 support examples per episode, offering reasonable coverage for mining common co-occurrences. We acknowledge that extremely low-shot settings (e.g., 1-shot) would increase sensitivity, as reflected in our shot-count ablation (Table 2).
>
> 2. Our model was designed to be able to perform in highly realistic scenarios while keeping the parameters low. While we could compare our model to large-scale pretrained models, that would defeat the purpose of a few-shot setting. We have also touched upon this in our limitations section in our paper
>
> 3. In our revised PDF, we have included standard deviations across multiple runs and are adding calibration metrics (AUPRC, macro/micro F1). We have also added a label-graph CRF baseline (~0.14 mAP), which our current model substantially outperforms. Regarding representativeness, prior few-shot multi-label audio methods use different evaluation protocols (base/novel splits with 100+ base classes for pretraining), making direct comparison infeasible without re-implementing under our harder all-class episodic setting.
>
> 4. We have looked at all the formatting issues in our paper, specifically \cite vs \citep, figure 1 not helping in any way, figure 2 not being informative enough, etc.
>
> 5. While yes, many recent works exist on few-shot and audio tagging, they do not touch our problem specifically. However, we have taken note and made a larger Related Work section to include all of them, and also explained why they don't touch our problem in another comment.
>
> Thank you once again, and hope this helped.

---

### Official Review · Reviewer_nHBQ · 2025-11-02

**Soundness:** 3
**Presentation:** 3
**Contribution:** 2
**Rating:** 6
**Confidence:** 3

**Summary:**

The authors present GRAFTa, a graph-based framework for few-shot multilabel
audio tagging. The model uses a CNN14 backbone to extract per-frame embeddings
from the input spectrograms. Then a graph attention network is used and both
single-class and compositional subset prototypes are utilized.

**Strengths:**

The proposed method achieves decent results on the FSD50K and Audioset datasets
(Section 6.1). Please also see the paragraph under Weaknesses.

**Weaknesses:**

The proposed method achieves decent results on the FSD50K and Audioset datasets
(Section 6.1). However, the CNN + ProtoNet comparison method achieves better
mAP performance. The authors point out that the per node dimension of GRAFTa is
64, versus 512 for the CMM + ProtoNet method. However, it seems that in both
cases the resource usage (e.g., GPU memory and training time) is quite limited.
The authors should discuss more what the benefits of GRAFTa are for practical
purposes. Maybe a comparison on mobile devises could be done?

**Questions:**

CNN + ProtoNet achieve a higher mAP score. What are the benefits of GRAFTa
beyond the lower per node dimension?

---

> ### Author Response · Authors · 2025-11-21
> **Correction of Experimental Setup and Updated GRAFTa Results**
>
> Thank you for reviewing our research paper. We will now try to answer your questions to the best of our abilities.
>
> 1) We identified that our original comparison used mismatched per-node dimensions (64 vs 512), unfair K values (GRAFTa performs better at higher K values), and we sent incorrect nodes to the GAT, which resulted in GRAFTa performing poorly. We have updated the  experiments with properly controlled settings, such as:
> - Made per node dimensions of both models to 256
> - Increased K values of both models to 256
> - Changed number of nodes (N) sent to GAT for both models from 4 to 33 because now we are taking the mean across the frequency dimension instead of time for temporal modelling which is better suited for GRAFTa.
>
> After making these changes, GRAFTa now outperforms CNN+ProtoNet. We have also added standard deviations across multiple runs. Here are the results below:
> FSD50K:
> GRAFTa - 0.2271 +- 0.0575
> CNN+Proto- 0.2056 +- 0.0441
>
> Audioset:
> GRAFTa - 0.2302 +- 0.0557
> CNN+Proto - 0.1936 + 0.0466
>
> 2)Beyond improved performance, GRAFTa offers two key advantages: (1) its graph-based architecture would be able to model spectrogram images better than CNNs due to non-local modelling, which would typically require more layers for CNNs to come close, and (2) compositional subset prototypes that capture label co-occurrence patterns, which are critical for multi-label few-shot settings, as shown in our ablation (Table 1).
>
> We are currently working on our revised paper and incorporating all the changes discussed above, along with additional feedback from the other reviewers. We’ll be uploading the revised PDF and supplementary materials within the rebuttal window. We wanted to respond quickly to acknowledge the reviewers’ concerns and share our new experimental results. We’ll post a follow-up comment once the full revision is ready. Thank you once again.

---

### Note · Program_Chairs · 2026-01-17
**Submission Desk Rejected by Program Chairs**

The following references in this submission do not refer to real documents and/or have major errors in bibliographic information:

 Yupeng Lu, Yingqiang Ge, John Hershey, Abdelrahman Mohamed, and Chao Wang. Spectrogramgraph: Spectrogram-based graph neural network for weakly supervised audio event detection. In IEEE International Conference on Acoustics, Speech and Signal Processing (ICASSP), pp. $911-915,2022$.